# Prospects for Data Collection to Optimise Kid Rearing in Dutch Dairy Goat Herds

**DOI:** 10.3390/ani15111653

**Published:** 2025-06-03

**Authors:** Eveline Dijkstra, Inge Santman-Berends, Tara de Haan, Gerdien van Schaik, René van den Brom, Arjan Stegeman

**Affiliations:** 1Department of Small Ruminant Health, Royal GD, Arnsbergstraat 7, 7418 EZ Deventer, The Netherlands; t.d.haan@gddiergezondheid.nl (T.d.H.); r.vd.brom@gddiergezondheid.nl (R.v.d.B.); 2Department of Research and Development, Royal GD, Arnsbergstraat 7, 7418 EZ Deventer, The Netherlands; i.santman@gddiergezondheid.nl (I.S.-B.); g.v.schaik@gddiergezondheid.nl (G.v.S.); 3Department of Population Health Sciences, Faculty of Veterinary Medicine, Utrecht University, Yalelaan 7, 3584 CL Utrech, The Netherlands; j.a.stegeman@uu.nl

**Keywords:** dairy goats, decision support tool, kid rearing, perinatal period, average daily gain, participatory approach

## Abstract

Good management of young dairy goats is essential for healthy and productive herds, but farmers often lack practical tools to monitor and improve rearing practices. This study actively involved farmers, veterinarians, and researchers to develop a set of measurable indicators related to early life care, such as birth weight, colostrum intake, growth, and kid survival. These indicators were then tested as proof of principle on five Dutch commercial dairy goat farms, with data collected from over 700 kids from birth to mating age. The results showed the added value of the collection of these data, revealing considerable differences between farms in how kids were raised and how they developed, particularly after weaning. Nevertheless, it also showed that the collection of the developed indicators was challenging, labour-intensive, and not always feasible, especially during the kidding season, which is the most labour-intensive period on a dairy goat farm. Farmers acknowledged the added value of the indicators but expressed a strong need for a digital tool to simplify data collection and interpretation. This study demonstrates that individual-level data can offer important insights into rearing performance and support more informed management decisions. Additionally, these findings highlight the potential of tailored, data-driven approaches to improve young stock care and support more sustainable dairy goat farming practices.

## 1. Introduction

The rearing period is crucial for production performance in dairy goats [1]. Effective rearing systems provide the foundation for raising the next generation of productive dairy goats. Goat kids are not only essential to initiate lactation but also play a pivotal role in herd replacement and genetic improvement. Furthermore, in the context of sustainable livestock farming, reducing antimicrobial use, and increasing animal welfare, the management of young stock has gained increasing importance [2]. Optimising kid-rearing results enhances survival rates while promoting the physiological development of the next-generation dairy goats [3,4,5]. However, rearing performance is influenced by multiple pre- and postnatal risk factors [6,7,8]. Additionally, management practices and husbandry systems can vary substantially between farms. To improve kid-rearing practices and ensure the long-term productivity of milking goats, a comprehensive and farm-specific approach to management and husbandry is essential. In particular, producers must possess a thorough understanding of their own rearing practices to identify and implement improvement [9].

The adoption of decision support tools is steadily increasing across livestock industries, becoming a crucial component in enabling producers to manage their livestock efficiently [10]. In the dairy cattle industry, the success of data-driven tools has been exemplified by a reduction in calf mortality on Dutch farms, where detailed rearing data reporting has increased producers’ awareness of the importance of effective rearing management practices [11,12]. Similarly, the benefits of recording farm-specific performance data and employing benchmarking systems are gaining recognition in dairy goat farming, contributing to improvements in health, profitability, and welfare [8,13]. However, the lack of reference data on kid-rearing practices limits producers’ ability to evaluate and optimise their results [7,14]. While monitoring and benchmarking systems hold great potential, their added value in dairy goat farming remains underexplored.

To gain a comprehensive understanding of rearing performance, it is essential to monitor parameters beyond mortality rates, such as growth rates throughout the rearing period [5]. A wide range of management factors, including nutrition, maternal condition, birth weight, colostrum, biosecurity, and housing, are known to influence the rearing performances of young stock [7,15]. Yet, research identifying predictors for successful rearing in dairy goat kids, especially in European husbandry systems, is scarce [13].

Dairy goat farmers are increasingly collecting valuable data on kid rearing, such as birth weight, colostrum intake, weight at weaning, and mating. When systematically collected and interpreted, these data hold the potential to support informed decision-making at both the herd and the individual levels. However, despite this promise, many farmers remain hesitant to adopt data-driven technologies because of a lack of engagement in their development. A top-down approach, with minimal farmer involvement during the early stages of development, has been identified as a major barrier to adoption [16].

Although data-driven tools have proven successful in improving young stock rearing in other livestock industries, their implementation in dairy goat farming remains limited, primarily due to the lack of reference data. As a result, there is a lack of understanding about the potential benefits of on-farm rearing data. In addition, it is unclear which predictors should be monitored and whether collecting data on these predictors is practically achievable. The hypothesis is that gaining insight into kid-rearing results supports improvement in on-farm rearing management, resulting in reduced kid mortality and optimal kid development.

Therefore, the aim of this study was to develop and evaluate key performance indicators (KPIs) to assess and optimise kid-rearing practices on dairy goat farms. The development of the prototype set of KPIs was achieved through a participatory approach that included input from scientists, veterinarians, and farmers. Following the creation of this prototype, the KPIs were assessed in terms of their benefits, feasibility, and potential barriers by collecting data from a select number of dairy goat farms as a proof of principle.

## 2. Materials and Methods

### 2.1. Study Design

To develop KPIs related to the quality of kid rearing, a participatory approach was adopted, involving both stakeholders (such as veterinarians and researchers) and end users (specifically farmers). The study engaged researchers specialising in small-ruminant health, veterinary epidemiologists, veterinarians, and five dairy goat farmers. As the study aimed to both develop KPIs and assess their added value for farmers, the herds of the participating farmers had to meet specific criteria. These included a willingness to actively engage in workshops, grant access to (routinely) collected data, and test the prototype KPIs on their farms during a single kidding and rearing period in 2020. An open call for participation was disseminated to all 400 Dutch dairy goat farmers via various (social) media channels. Five farms were selected that had more than 500 dairy goats, had a scheduled kidding period between June and August 2020, and were officially disease-free from caseous lymphadenitis (CL) and caprine arthritis and encephalitis (CAE).

### 2.2. Development of Key Performing Indicators Related to Kid Rearing

Prior to data collection, three workshops were held to align objectives and foster collaboration. In the first workshop, researchers and five dairy goat farmers conducted a plenary brainstorming session using a “post-it” exercise. Participants listed parameters indicative of kid performance, which were then categorised into themes such as birth and growth. Through group discussion, a consensus was reached on key parameters considered valuable and feasible for on-farm data collection. The full list of discussed parameters is provided in Appendix A.

The second workshop involved small ruminant veterinarians and veterinary epidemiologists. Discussions focused on defining each parameter, assessing its measurability, and evaluating feasibility based on four criteria: (1) direct relevance to kid rearing and health; (2) applicability across goat farms; (3) availability in existing farm records; and (4) practicality of on-farm recording. Subsequently, four distinct rearing phases of interest were defined.

In the third workshop, all stakeholders finalised the KPI list. Parameters were prioritised based on their relevance to knowledge gaps identified by farmers and the feasibility of individual-level data collection, as evaluated by farmers, veterinarians and epidemiologists. The group also defined practical requirements for high-quality data collection, including farmer support, availability of data recorders during peak kidding, and user-friendly registration tools (available in Dutch from the first author upon request). Based on the outcomes of this workshop, the prototype list of KPIs was created.

After completion of the feasibility assessment during the proof of principle in which data of the selected KPIs were collected during a kidding season, a fourth and final workshop was organised in which farm-specific results were shared individually, and anonymised group findings were presented and discussed collectively. This final session allowed for joint interpretation of results and evaluation of the data collection process, offering valuable insights into the added value, the barriers, feasibility and potential of a tailored data collection and analysis tool for improving rearing management.

### 2.3. Assessment of Feasibility of the Defined Key Performance Indicators for Kid Rearing

For each defined KPI, an assessment was conducted to determine whether data were routinely collected and already available on the farms. Since 2010, producers have to register all goats present on the farm in the national identification and registration database (I&R), managed by the Dutch enterprise agency (Rijksdienst voor ondernemend Nederland, Assen, The Netherlands), in compliance with Council Regulation (EC) No. 21/2004 [17]. Subsequently, it was determined which additional data needed to be actively collected on-farm. In consultation with the farmers, standardised recording sheets were developed to record the data not routinely available. Sheets were available for each of the four kid-rearing phases and were distributed to ensure uniform data collection across the five farms. During the data recording period, producers were regularly consulted to identify any difficulties encountered in data collection. Where farmers indicated the need for assistance, personal support was provided to ensure accurate and complete registration of data. All completed recording sheets were submitted to the researchers for analysis.

The added value of selected KPIs was assessed on the variability between and within farms, as well as farmers’ feedback on whether the benefits justify the labour involved in data registration. Data collected from the participating farms were descriptively analysed. Summary statistics (i.e., means, medians, and variation among kids) were calculated for the postnatal, postweaning, and final rearing periods, with results presented in tables and figures. Differences between farms were assessed using the appropriate tests: ANOVA for normally distributed parameters and the Kruskal-Wallis test for non-normally distributed parameters.

Where applicable, associations between rearing management and the dependent variable were assessed using a univariate multilevel linear regression model, incorporating a random farm effect. Statistical analyses were performed in Stata version 17^®^. When necessary, parameters were log-transformed to meet the assumptions of normality according to the skewness and kurtosis test in Stata 17 (procedure: sktest). Normality was evaluated by visual inspection of Q-Q plots of residuals and scatterplots observed against fitted values. Variables potentially associated with higher weight gains were identified based on a significance threshold of *p*-value ≤ 0.05.

## 3. Results

### 3.1. Prototype Set of Key Variables

Based on the input from stakeholders and end users in the development phase, the relevant rearing period was defined as the time from birth to mating (approximately seven months of age). A prototype set of KPIs for monitoring kid rearing was identified and categorised into the four defined rearing phases: perinatal (first 48 h), postnatal (birth to weaning), postweaning (weaning to 12 weeks), and final rearing (12 weeks to mating) (Figure 1). Each KPI was calculated at the individual level and subsequently averaged at the herd level. Where relevant, a benchmark across farms was added. The final set of prototype KPIs selected for the on-farm testing phase involved (1) birth, (2) colostrum management, and (3) growth during each rearing phase. Mortality was assessed in two periods, from birth to 21 days of age (KPI 4) and from birth to 91 days (12 weeks of age; KPI 5). Growth parameters for each phase were presented as average daily gain (ADG) and were calculated as outlined below.

Average daily gain from birth to time of mating (g/day), taking into account the number of days between the date of birth and the date the mating weight was determined.


$${ADG}_{tot}=\frac{{Weight}_{t_{mating}}-{Weight}_{t_{birth}}}{n\;days}$$


2.Average daily gain during the postnatal period (ADG_postnatal_): the difference in weight between the dates of weaning and birth, using the number of days between the two weighing moments as the denominator.3.Average daily gain during the postweaning period (ADG_postweaning_): the difference in weight between weaning and the 12-week weighing, using the number of days between the two weighing moments as the denominator.4.Average daily gain during the final period (ADG_final_): the difference in weight between 12 weeks and mating, using the number of days between the two weighing moments as the denominator.

To enable the calculation of the selected KPIs, detailed data were required on birth, colostrum, weaning, mating, and weights (Figure 1). To minimise additional labour involved in on-farm data collection, herd-specific data were retrieved from the I&R system where possible. These data included the unique herd identification number (UHI), unique animal ID, date of birth, and the date and reason for off-farm movement (e.g., movement to another farm to slaughter or mortality). Additional data needed for KPI calculation required on-farm data collection. On-farm records comprised data gathered at, or shortly after, birth, including the date and time of birth, the sex and birth weight (g) of the kid, and the parity and litter size of the doe. For each colostrum feeding event, the date and time of feeding, source (goat, artificial, or a combination), volume administered (mL), colostrum quality (Brix value), and method of feeding (bottle, tube, or other) were recorded. At weaning, both the date of weaning and weaning weight were recorded, followed by weight measurements at 12 weeks and just before mating. This procedure covered a six-month rearing period for all kids born between June and August 2020.

### 3.2. Outcomes of the Assessment of Feasibility and Added Value of Selected KPIs in Practice

Data were collected on goat kids born between June and August 2020, with the final data obtained in January 2021. The average farm size of the five herds was 1004 goats (>1 year), with sizes ranging from 530 to 1507. White Saanen was the predominant breed across all herds, and none of the five farms practised grazing. The quantity and quality of data varied considerably among the five farms (A–E), as detailed in Table 1.

Births were recorded at all farms; however, a cross-check with the I&R database revealed that Farms C and D primarily recorded births of doelings. Farm D weighed kids frequently but at random moments rather than at the agreed-upon time points. Additionally, despite frequent contact, none of the other predetermined variables were sufficiently recorded by this farmer. Farm E returned incomplete and low-quality data sheets, which hindered follow-up on individual animals. Due to the missing and unreliable data, farms D and E were excluded from further analysis to ensure the presentation of high-quality results and meaningful comparisons. Across Farms A, B, and C, 816 kid births were recorded in the I&R database. However, Farm C recorded data for only 338 of the 439 kids born (77%). Consequently, the dataset used for further analysis comprised 715 kids born between 29 May and 6 July 2020, including 416 doelings (58%) and 299 buck kids (42%) (Table 1).

### 3.3. Perinatal Period

Most kids weighed between 3.0 and 3.5 kg across all three herds, with a median birth weight of 3.5 kg. Buck kids had a median weight of 3.7 kg (*n* = 285), while doelings had a median weight of 3.3 kg (*n* = 399) (Figure 2; Table 2). Farms A and C both reported a median birth weight of 3.5 kg, whereas Farm B had a significantly lower median of 3.2 kg (*p* < 0.05) (Figure 2; Table 2). The distribution of kid weights at Farm B was skewed towards lower values, while Farm C showed a tendency towards higher weight values (Figure 2). When corrected for sex, buck kids weighed significantly more than doelings within each farm (*p* < 0.05).

Dams of the included kids were, on average, in their second parity and delivered mean litter sizes of 2.4 kids (Table 2). Across all farms, kids were separated from their dams directly after birth and received their first colostrum approximately one hour postpartum. Median intervals between birth and first colostrum administration ranged from 35 to 105 min across farms, with no substantial variation observed. Farm A recorded additional data for 115 kids, showing an average interval of nearly 12 h between the first and second colostrum feedings. On this farm, the second feeding averaged 188 mL. Total colostrum intake was assessed across all three farms, with an average of approximately 500 mL per kid.

However, differences between farms were evident: Farm C provided the highest average total colostrum volume, about 100 mL more than Farm B, which recorded the lowest intake. This pattern was also reflected in the percentage of colostrum relative to birth weight. Farms A and C administered colostrum equivalent to approximately 16% of birth weight compared with 13% at Farm B. Goat colostrum was the most frequently used type of colostrum across all three farms (74%), while artificial colostrum was used only sporadically at Farms A (1%) and C (2%). In contrast, Farm B fed 12% of its kids with artificial colostrum.

Feeding methods also varied between farms. Tube feeding was the standard practice at Farm A (79%), while Farm B exclusively used bottle feeding. Farm C primarily used bottle feeding (73%), with tube feeding accounting for 25% of colostrum administrations.

### 3.4. Postnatal and Postweaning Period

In all three herds, most buck kids were moved off-farm at one week of age. Consequently, growth parameters were based almost exclusively based on data from doelings. The mean weaning weight across the three farms was 15.4 kg, although both weaning age and weight varied considerably between farms (Table 3). Kids were weaned at a median age of 58 days on Farm C, compared with 65 and 67 days on Farms A and B, respectively. Growth rates during the postnatal period showed considerable variation between farms. Farms A and B reported a median daily weight of approximately 170 g, whereas Farm C achieved a notably higher median gain of around 216 g per day (Table 3).

At the 12-week weighing, kids weighed on average 17.3 kg, with a median interval of 21 days postweaning. Across the three farms, the average weight gain during the postweaning period was 133 g per day (median 129 g/day). Although Farm A had the lowest weaning weights, it reported the highest 12-week weights compared with Farms B and C (Figure 3). Remarkably, kids on Farm C experienced a significant drop in ADG to 89 g per day between weaning and 12 weeks of age. In contrast, such a decline was not observed in Farms A and B, which reported gains of 203 and 147 g per day, respectively (Figure 3). Furthermore, the 12-week weighing revealed the greatest variation in kid weights across all three farms.

Mating weights were, on average, recorded 114 days (median 110) after the 12-week weighing. At mating, doelings weighed on average 36.1 kg, equating to an ADG of 174 g per day (median 175 g/d) from 12 weeks of age to mating. Farm B recorded the lowest median mating weights, although growth rates during the final rearing period were similar across all farms, with ADGs of 187, 168, and 176 g per day on Farms A, B, and C, respectively.

Survival and mortality data were recorded for 667 kids. Nineteen kids died within the first 21 days of life, corresponding to a mortality rate of 2.8% during this period. An additional 13 deaths were recorded between 22 to 91 days of age (Table 1), resulting in an overall mortality rate of 4.8% within the first 91 days postpartum. Mortality rates varied between farms: within the first 21 days, rates were 4.7%, 5.3%, and 0.0% for farms A, B, and C, respectively. By 91 days of age, these rates had increased to 5.4%, 5.7%, and 3.3%, respectively.

### 3.5. Associations Between Rearing Management and Average Daily Weight Gain

Birth weight was positively associated with weight gain. Kids with higher birth weights gained significantly more weight between birth and mating compared with those with lower birth weights (*p* < 0.001; Table 4). Additionally, results showed that a higher weaning weight was associated with faster growth between 12 weeks and mating (*p* < 0.001). A higher total colostrum intake tended to be associated with a higher weight gain across the total rearing period (*p* = 0.13). However, colostrum intake volume was correlated with birth weight (r = 0.43).

Kids that gained weight quickly during the postnatal period were weaned at an earlier age, consequently leading to a longer postweaning period. Colostrum quality, measured using Brix values, was recorded only at Farm C. Higher Brix values were significantly associated with a higher weight gain throughout the total rearing period (*p* = 0.03). Feeding methods also affected growth performance, with tube feeding of colostrum showing a positive association with weight gain during the final rearing period compared with bottle feeding (*p* < 0.001; Table 4). The proportion of kids fed artificial colostrum was too low to evaluate its association with weight gain.

## 4. Discussion

Emerging in the 1980s, the Dutch dairy goat industry remains relatively young and continues to face important knowledge gaps, particularly in areas like young stock rearing. Managing large herds, averaging 1300 dairy goats and 800 births annually [8], poses considerable challenges, especially given the complexity of balancing various production demands with seasonal workloads. Sustainable dairy goat farming requires a dual focus on optimising individual kid development and maintaining effective herd-level monitoring. However, high workloads might hinder detailed individual data collection, which is crucial for understanding and improving rearing practices. Encouraging farmers to collect data depends on demonstrating its value relative to the effort involved [12,18]. This study aimed to bridge this gap by addressing two central objectives: develop a set of KPIs to provide insight into the quality of kid rearing using a participatory approach and, second, to establish a proof of principle by testing the feasibility and practicality of collecting individual data for calculation of the developed KPIs across five commercial dairy goat farms. Engaging farmers in the process was central to this approach, as their involvement in the development of the KPIs fostered practical relevance and increased the likelihood of adoption [16,19]. By demonstrating the value of such data for improving rearing outcomes and farm management, this study provides an important step toward integrating systematic data collection into routine farming practices.

Building on the participatory approach outlined above, this study focused on collecting high-quality data on kid rearing across four distinct rearing stages: birth, weaning, 12 weeks of age, and mating. During the preparatory workshops, farmers expressed a clear need for support in visualising and understanding rearing outcomes at each of these stages. Although five farmers were actively involved during the design phase, complete data were ultimately obtained from only three farms. Nevertheless, according to both researchers and participating farmers, a valuable and unique data set with a benchmark was obtained, comprising individual records from 715 kids monitored from birth to mating.

The proof of principle analysis highlighted both the challenges but also the value of farm-specific data. Considerable variability in both collected variables and outcome parameters was observed across farms. Remarkably, despite the limited number of participating farms, significant inter-farm differences emerged, particularly in weight and growth-related parameters. Weight gain in kids, influenced by factors such as colostrum management, kid environment, disease, and nutrition, is considered an important indicator of productivity in adulthood [20]. The observed variation in growth outcomes underscores the significant impact of farm-specific management practices [14,21]. While the overarching concept of rearing goat kids did not differ largely between farms, notable differences in management choices, including housing, staffing, colostrum management, feeding, weaning strategies, and preventive measures, were likely responsible for the observed disparities in rearing performance.

Birth weights appeared to be one of the best-recorded variables in this study. The results consistently showed that birth weight was positively associated with all growth parameters except during the final rearing period. Higher birth weights were significantly correlated with shorter intervals between birth and weaning, which can be attributed to higher weight gain observed in heavier kids at birth. For multiple ruminant species, birth weight is regarded as an important predictor for subsequent growth [22]. Across all three farms, birth weights varied widely, ranging from 2.1 to 5.0 kg, with an average of 3.5 kg. Ideally, birth weights lie between 3 and 5 kg [1,21]. Importantly, the distribution of birth weights was significantly different between farms. Consistent with previous studies, buck kids weighed more than doelings [23,24]. When analysing birth weights as a predictor for farm performance, it is essential to calculate average weights separately for buck kids and doelings, as this distinction can reveal important trends. Kids with birth weights below 3 kg are at greater risk of rearing challenges and reduced survival rates [1]. Low mean birth weight should prompt targeted preventive measures such as improved colostrum management, enhanced monitoring and tailored nutritional strategies. To increase birth weight, management practices for pregnant does must be carefully evaluated. Key areas for consideration include maintaining body condition, ensuring adequate nutrition during gestation, selecting favourable breeding genetics, mitigating disease risks, and providing stress-free housing to support foetal development [25,26]. Addressing these factors could significantly enhance birth weights and subsequent rearing success across farms.

Directly after birth, goat kids rely entirely on colostrum to establish passive immunity [27,28]. Effective colostrum management, encompassing type and quality, timing, frequency, volume, and feeding systems, is critical for kid development and is often evaluated during rearing challenges [21]. In this study, the first colostrum was, on average, provided 60 min postpartum, ranging from 10 and 160 min, and all farms exceeded the recommended colostrum intake of 10% of birth weight within 24 h [1]. Feeding methods varied. Farm A used a stomach tube and recorded the timing of the second colostrum feedings, whereas Farms B and C relied primarily on bottles and recorded the total amount of colostrum. When applying a suckling method, more frequent and smaller portions of around 150–200 mL/kg BW are recommended [1]. The authors cannot explain the significant association between tube feeding and total growth. No significant associations between the type of colostrum and growth were found, likely because all three farms mainly used goat colostrum. Farm C recorded median colostrum Brix values of 24.1 mg/mL, exceeding the ≥20 mg/mL cut-off for good quality colostrum [29,30]. On this farm, higher Brix values were positively associated with weight gain, consistent with other studies [14,31]. Colostrum provides not only immunoglobulins but also essential growth hormones, fats, and other nutrients that support the early development of kids [32].

The mean weaning weight across farms was 15.4 kg, aligning with reference values for large breeds [21]. Differences in median weight gain between farms were most pronounced postweaning, consistent with the results of Deeming et al. [33] Within each farm, the greatest variations in growth parameters were observed during the postweaning period. Weaning age and weaning weight varied between farms, reflecting the use of weight-based criteria. Farm A aimed to wean at 13 kg, while Farm B and C aimed at 14 kg for females and 16 kg for male kids. A consequence of weight-based weaning is that the weaning age depends on growth performance. High ADG causes kids to be weaned at a younger age, and between birth and weaning, Farms A and B had an approximate ADG of 170 g/day, whereas kids at Farm C gained 216 g/day. On the latter farm, kids were weaned generally a week earlier compared with the other farms. However, the postweaning ADG on Farm C dropped dramatically to 86 g/day compared with more stable postweaning gains at Farms A and B. The timing and method of weaning are critical for gastrointestinal development and growth. Early weaning risks underdevelopment of the gastrointestinal tract, while older weaning ages promote better adaptation of bacterial flora and improved digestive efficiency [34]. The postweaning growth shock on Farm C, however, might also have been the result of the applied weaning method. At this farm, kids received ad libitum milk until weaning. Farms A and B gradually reduced milk intake from six weeks of age until weaning. Gradual weaning strategies are found to stimulate solid feed intake before weaning and diminish the impact of weaning on ADG postweaning compared to sudden weaning [1,35,36].

Participants were asked to register mortality data during the rearing period, though stillbirths were not recorded. This resulted in average mortality percentages of 2.8% and 4.8% over the period from birth to 21 days and from birth to 91 days, respectively. However, these low rates, combined with the limited number of farms studied, restricted the assessment of associations between rearing management and mortality. The recorded mortality likely underestimated the actual rates, as most farmers only recorded mortality of ear-tagged kids, who were generally over one week old. One farmer recorded no mortality within the first 21 days postpartum, and male kids were not included. Incomplete mortality data can skew rearing performance assessment, potentially leading to an overestimation of average growth and masking associations between management practices and successful kid-rearing. However, recent amendments to the mandatory identification and registration of goat kids are expected to improve the quality and completeness of mortality data collection. Since November 2020, Dutch dairy goat farmers have been obligated to register kids within seven days postpartum, report stillbirths, and specify the gender of each kid [37].

Although the added value of individual data collection was evident, the results of the proof of principle also clearly demonstrated that recording such farm-specific data remains inherently challenging [6], with successful implementation depending strongly on farmers’ motivation and perceived value of the data. A participatory approach was adopted to enhance engagement by incorporating farmers’ perspectives into the development of the KPIs and the design of the data collection process. However, despite this involvement and provision of intensive guidance, two of the five participating farmers failed to return complete and good-quality datasets. The farmers cited that the perception of the significant workload during the kidding period was a challenge and that motivation alone was not always sufficient to overcome practical barriers. Notably, data collection during the first 48 h postpartum was deemed the most feasible to integrate into existing working routines. This might suggest that farmers are more likely to engage in data recording when it is considered manageable and can be easily incorporated into routine work procedures. All farmers acknowledged the value of extended rearing data for on-farm management. The data collected provided detailed insights into within-farm and between-farm variability, supported the evaluation of management consistency, and helped identify critical rearing phases where deviations in outcomes occurred. Feedback sessions sparked a lively discussion among the farmers regarding the observed differences between herds. These discussions underscored the potential of shared learning and benchmarking to enhance management practices. A further challenge identified in this study was inconsistency in data collection. Not all farmers recorded every variable agreed upon during the development phase, often reflecting the individual assessment of the perceived added value against the effort required. To support broader implementation in future initiatives, it will be essential to develop clearly defined, standardised, and practically applicable data collection protocols. While intensive support during this study was valuable in facilitating adoption, such guidance is unlikely to be sustainable at a larger scale. Ensuring that data collection methods are reproducible and rigorous, yet straightforward to implement independently by farmers, is therefore essential. Looking ahead and based on discussions with the farmers, the development of a digital tool to support data collection was deemed necessary to make the collection of individual animal data feasible and sustainable. All five farmers expressed interest in a solution that could provide clear insights into kid-rearing practices, including benchmarking features to establish comparative baselines [13]. They agreed that such a tool should be digitalised and designed with multiple-choice options or numerical inputs to minimise the additional workload. A tiered structure may offer a promising solution to accommodate both less and more engaged farmers by allowing them to choose the level of data entry and analysis that suits their capacity and interest, offering a basic version that primarily relies on routinely collected data, with optional expansions for more motivated users interested in recording additional variables. Farmers also emphasised the importance of tools that do not merely collect data but provide tangible value for management. Developments in smartphone technology, access to mobile internet, and cloud services make the development of a smartphone application to support farmers’ decision-making feasible [38]. Such an app could streamline data collection by integrating automated tools like electronic ear tag scanners to record individual variables (e.g., animal ID, birth weight, colostrum quality, and ADG) and improve on-farm data organisation. Ideally, the tool would allow real-time data analysis and feedback, further enhancing its practical utility and relevance for daily herd management.

While collecting data inevitably adds to the farmers’ workload, those already engaged in record-keeping often demonstrate higher rearing quality and better overall farm performance [14]. This suggests that record keeping not only reflects good farm management but also provides valuable information that likely supports effective decision-making, thereby enhancing animal performance. To encourage broader adoption, the development of a user-friendly digital tool is essential. The tool must provide detailed, comprehensible insights into herd-specific kid-rearing metrics and offer actionable feedback to dairy goat producers. By addressing these needs, such innovations could support improved management and rearing outcomes while fostering a culture of continuous learning and data-driven decision-making in the dairy goat industry.

## 5. Conclusions

Collecting data at the individual animal level has significant added value for the optimisation of kid-rearing management. Among the various rearing variables, mortality and weight data are considered the most relevant. Consistently recording these variables should serve as the foundation of any data management tool, as they provide the context needed to interpret additional variables if the dataset is expanded. However, collecting such data is challenging for farmers. Even highly motivated farmers struggle to gather this information during the kidding period, which is the most labour-intensive time of the year. To address these challenges and ensure the collection of complete, high-quality data, any future tool must prioritise automation. By integrating automated processes, the tool can reduce the burden on farmers while maintaining data integrity. In this way, the combination of automation and practical utility can drive sustained compliance and improve management practices in the dairy goat industry. Furthermore, this study has demonstrated the importance of incorporating the proof of principle phase in research of this nature, as field validation provides critical insight into both the practical challenges and the added value of data collection, which emphasises the necessity of field validation when aiming to develop effective, implementable tools for farm management.

## Figures and Tables

**Figure 1 animals-15-01653-f001:**
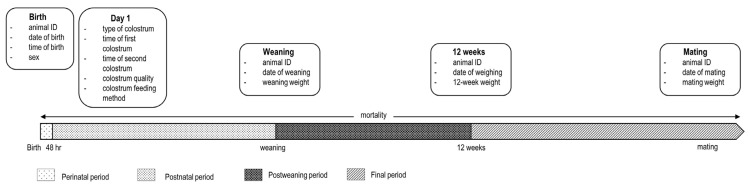
Visual representation of the individual-level data recorded by dairy goat farmers between June 2020 and January 2021.

**Figure 2 animals-15-01653-f002:**
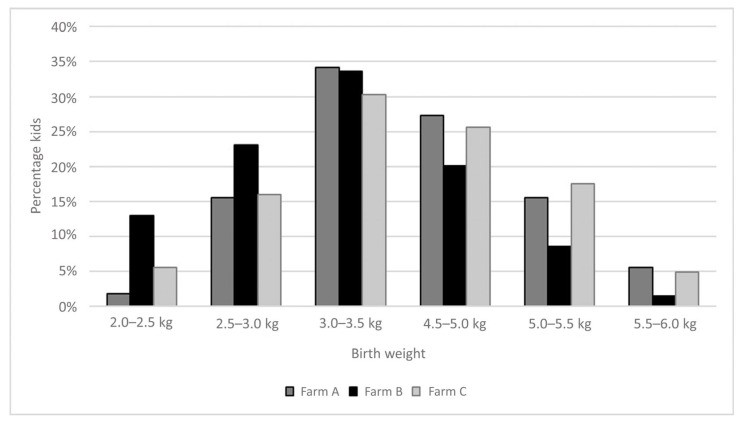
Distribution of recorded birth weights in kg for Farms A, B, and C. Median birth weight differed significantly between farms (*p* < 0.05).

**Figure 3 animals-15-01653-f003:**
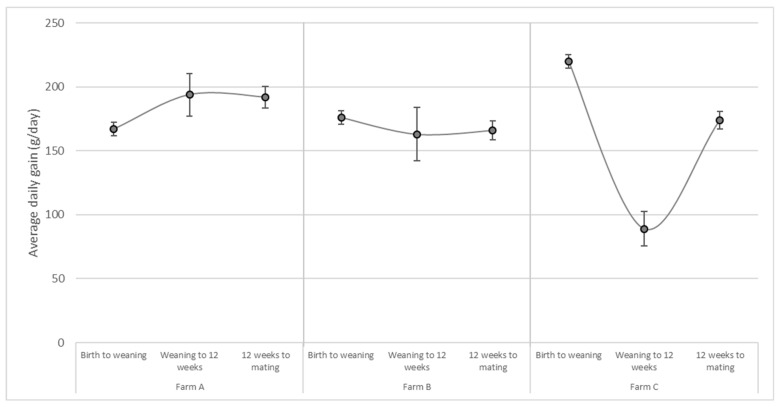
Average daily weight gain in grams/day across three rearing phases in goat kids from three dairy goat farms between June 2020 and January 2021.

**Table 1 animals-15-01653-t001:** Number of kid records per parameter per farm during the study period from June 2020 to January 2021.

Variable	Number of Recordings per Farm				
	A	B	C	D	E
Number of milking goats (>1 year)	544	1210	1227	1507	530

No. of births (I&R)	168	209	439	211	180
Doelings as registered by the farmer	72	109	235	206	62
Bucks, as registered by the farmer	96	100	103	5	57
Unknown	0	0	101	0	61

No. of birth weight records	161	199	324	-	95

No. of weaning weight records	58	87	173	-	21

No. of weight at 12 weeks records	56	103	146	-	-

No. of mating weight records	60	109	190	-	-

Registered mortality of kids	11	14	11	2	32
≤21 days	8	11	0	0	1
22–91 days	1	1	11	0	25
>92 days	2	2	0	2	6

**Table 2 animals-15-01653-t002:** Summary of colostrum management practices in three Dutch dairy goat farms between June and August 2020.

Parameter	N	Mean	Median	P5 *	P95 *	Median per Farm		
						A	B	C
Birth weight (kg)	684	3.4	3.5	2.5	4.5	3.5	3.2	3.5
Parity dam	429	1.8	2	1	3	-	1	2
Litter size	222	2.4	2	1	4	-	-	2
Time 1: birth to 1st colostrum (min.)	582	66	60	15	158	35	105	60
Time 2: 1st and 2nd colostrum (min.)	115	719	725	270	1125	725	-	-
Colostrum quality (Brix value)	313	24.4	24.1	17.7	33.2	-	-	24.1
Colostrum intake (mL)	615	504	500	300	725	538	425	550
Colostrum to birth weight ratio (%)	590	15%	15%	8%	21%	16	13	16
Type of colostrum						**Percentage per farm (%)**		
Goat	529					79	45	89
Artificial	36					1	12	2
Combination	150					20	43	9
Method of administration								
Tube	250					98	0	25
Bottle	458					2	100	73
Combination	7					0	0	2

* P5 = 5th percentile, P95 = 95th percentile.

**Table 3 animals-15-01653-t003:** Summary of growth variables and parameters recorded by Farms A, B, and C between May 2020 and January 2021.

Parameter	N	Mean	Median	P5	P95	Median per Farm		
						A	B	C
Birth weight (g)	684	3445	3450	2480	4450	3500	3200	3500
Weaning weight (kg)	318	15.4	15.2	13.2	18.6	13.6	14.6	16.0
12-week weight (kg)	306	17.3	17.0	13.9	22.0	19.0	15.8	17.4
Mating weight (kg)	359	36.1	37.0	21.5	46.5	39.1	34.5	38.0
Time postnatal period (days)	330	62	61	48	76	65	67	58
Time postweaning period (weeks)	286	17	21	7	29	21	12	21
Time final period (days)	295	114	110	104	137	109	111	110
ADG * postnatal (g/day)	328	199	195	142	270	164	174	216
ADG * postweaning (g/day)	284	133	129	−17	300	203	147	86
ADG * final period (g/day)	281	174	175	115	227	187	168	176

* Average daily weight gain.

**Table 4 animals-15-01653-t004:** Associations between rearing management practices and average daily weight gain of goat kids born between June and August 2020 on three dairy goat farms, using univariable multilevel linear regression models.

Parameter		ADG Total Rearing Period *		ADG Postnatal Period *		ADG Postweaning Period		ADG Final Period	
		Estimation (95% CI)	*p*	Estimation (95% CI)	*p*	Estimation (95% CI)	*p*	Estimation (95% CI)	*p*
Time to first colostrum (hr)		−0.01 (−0.03–0.15)	0.44	−0.01 (−0.03–0.009)	0.25	−6.01 (−21.9–9.9)	0.46	2.26 (−4.6–9.1)	0.52
Time between 1st and 2nd colostrum (hr)		−0.002 (−0.01–0.01)	0.65	0.002 (−0.005–0.009)	0.62	1.92 (−1.75–5.59)	0.31	−0.36 (−2.4–1.7)	0.73
Birth weight (100 g)		0.008 (0.005–0.01)	<0.001	0.004 (0.001–0.007)	0.006	2.72 (0.83–4.62)	0.005	0.33 (−0.44–1.09)	0.41
Weaning weight (100 g)		0.004 (0.003–0.005)	<0.001			−0.42 (−1.09–0.24)	0.21	0.49 (0.21–0.78)	0.001
Colostrum quality (Brix value)		0.006 (0.001–0.12)	0.03	0.002 (−0.002–0.007)	0.4	3.13 (0.21–6.05)	0.04	0.41 (−1.11–1.93)	0.6
Amount first colostrum (per 100 mL)		0.003 (−0.01–0.18)	0.67	0.009 (0.007–0.03)	0.27	−4.29 (−14.0–5.50)	0.39	unstable	
Total colostrum amount (per 100 mL)		0.01 (−0.003–0.26)	0.13	0.01 (0.004–0.02)	0.15	−2.45 (−11.0–6.09)	0.57	0.01 (−3.43–3.46)	0.99
Parity dam		−0.003 (−0.03–0.02)	0.76	0.009 (−0.12–0.30)	0.39	−0.50 (−13.1–12.1)	0.94	unstable	
Litter size (*n*)		−0.01 (0.04–0.02)	0.45	−0.002 (−0.02–0.02)	0.87	−5.9 (−20.1–8.4)	0.42	−0.33 (−0.02–0.02)	0.87
Time postnatal period (days)		−0.05 (−0.07–−0.04)	<0.001	−0.08 (−0.09–−0.06	<0.001	−4.28 (−15.3–6.7)	0.45	−0.08 (−7.5–6.9)	0.93
Time postweaning period (weeks)		0.007 (0.005–0.01)	<0.001			−0.56 (−2.31.2)	0.54	unstable	
Colostrum method (relative to tube)									
	Bottle	−0.03 (−0.08–0.01)	0.17	−0.02 (−0.07–0.03)	0.34	−21.8 (−53–9.4)	0.17	−16.7 (−26–−7.5)	<0.001
	Combination	−0.1 (−0.26–0.07)	0.25	−0.13 (−0.28–0.02)	0.09	−66.8 (−235–101)	0.44	−43.7 (−113–26)	0.22

* Log transformed, CI = confidence interval.

## Data Availability

The data supporting the findings of this study are available upon reasonable request. Interested researchers may contact the corresponding author for further details on data access and conditions for use.

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
