# Peer review of "Prospects for Data Collection to Optimise Kid Rearing in Dutch Dairy Goat Herds"

_animals, 2025, doi:10.3390/ani15111653_

Round 1

Reviewer 1 Report (Previous Reviewer 1)

Comments and Suggestions for Authors

The authors have made the suggested changes, the manuscript has improved considerably. I suggest its publication in the present form.

Reviewer 2 Report (Previous Reviewer 3)

Comments and Suggestions for Authors

Thanks for your review. I think that the manuscript can proceed with publication.

Reviewer 3 Report (Previous Reviewer 4)

Comments and Suggestions for Authors

Dear Authors,

Thank you very much for accepting my recommendations. The manuscript has been improved and could be accepted for publication. On the other hand, I still do not feel that the article is a scientific publication of the same level and value as the esteemed Authors stated in their response. Nevertheless, I congratulate them on their work!

This manuscript is a resubmission of an earlier submission. The following is a list of the peer review reports and author responses from that submission.

Round 1

Reviewer 1 Report

Comments and Suggestions for Authors

This research provides interesting information. However, it is necessary to make some important changes before final publication.

Simple Summary: adjust this section because it summarizes the implications of this research in a simple and informative way.

Abstract: Adjust this section. The main results, conclusions and implications of this research should be mentioned in a summarized manner.

MATERIAL AND METHODS

General comments:

Line 104.- they mention “Five farms” this was already mentioned on line 100. I recommend sorting out the ideas.

Line 202.- were the assumptions of normality (normal distribution and homogeneity of variances) evaluated in the residuals of the response variable? Also, I consider that the “P-value of ≤0.10” is changed to a “P-Value of ≤0.05”. It is too high this value. Therefore, change the significance level in the results section.

RESULTS

General comments:

I recommend restructuring this entire section with a significance of “P-Value of ≤0.05”.

Line 248-. They mention “The weight distribution of kids on farm B tended to be skewed toward lower weights while the weight distribution on farm C tended to be skewed toward higher weights (Figure 1).” Fig 1, says “Figure 1. Graphical representation of data collected by Dutch dairy goat farmers for each individual kid between June and August 2020”. They are referring to fig 2? Also, they mention “weight distribution on farm C tended”, specify what they mean by this.

Line 251.- Specify the statistical addresses by “Brith weight” and “Farm”.

Line 271.- Table 2, the “Mean” is decimated, I recommend mentioning “Mean±DE”. This should also be adjusted in table 3.

Line 313.- mention “Birth weight was positively associated with weight gain”. Specify how these data were analyzed in the “Data analysis” section.

DISCUSSION

In general, I recommend restructuring this section and explaining the findings observed in the research and trying to explain the mechanism involved.

Line 336-347.- support this paragraph with references.

Line 357-362.- support this paragraph with references.

Line 424-426.- mention “In other ruminant species, parity has been associated with colostrum IgG concentration [28], [35], and birth weight [36]”. You did not evaluate the “IgG” so you cannot relate it because it can be speculative. I recommend just focusing on the results of your research.

CONCLUSION

I recommend restructuring this section and being more specific about the findings mentioned in the results.

Comments on the Quality of English Language

REVIEW THE WRITING 

Author Response

Response to Reviewer 1

We sincerely thank you for your valuable suggestions and constructive feedback on our manuscript. Your comments highlighted that the aim of the article was not always clearly articulated and that structural improvements were necessary to better guide the reader through the objectives of the study.

In response, we have undertaken substantial revisions to enhance the clarity, coherence, and logical flow of the manuscript. Particular attention was given to clearly stating the study's aims to develop and evaluate key performance indicators (KPIs) to assess and optimise kid-rearing practices on dairy goat farms, and restructuring the manuscript to better align with and support this objective. To strengthen the narrative, we moved outcomes from the development phase, originally included in the Materials and Methods section, to the Results section. This adjustment clarifies the progression through the study’s three key phases: development, proof of principle in practise, and evaluation, and illustrates how each contributes to the overarching research aim. Additionally, the entire manuscript has been thoroughly reviewed and revised for grammar and language to improve overall readability and consistency.

We are very grateful for the time and effort you have taken to review our manuscript. Please find our detailed responses below, with the corresponding revisions and corrections highlighted in the re-submitted files using track changes.

Kind regards,
The authors

Point-by-point response to Comments and Suggestions for Authors

This research provides interesting information. However, it is necessary to make some important changes before final publication.

Simple Summary: adjust this section because it summarizes the implications of this research in a simple and informative way.

Abstract: Adjust this section. The main results, conclusions and implications of this research should be mentioned in a summarized manner.

We thank the reviewer for the positive evaluation of our research and for highlighting the need for improvements.

Simple Summary:
We appreciate your observation that the ‘Simple Summary’ required adjustment to more clearly and informatively convey the implications of our research. In response, we have revised this section to provide a concise and accessible overview of the study’s aim, in accordance with the journal’s  guidelines for a non-specialist audience.

Abstract:
Following your recommendation, we have revised the ‘Abstract’ to more effectively highlight the main results, conclusions, and implications of our work. The revised version now clearly outlines the research objective, methodology, and key findings, including variation in kid rearing practices, associations between rearing indicators and growth performance, and the practical relevance of the developed KPIs.

MATERIAL AND METHODS

General comments:

Line 104.- they mention “Five farms” this was already mentioned on line 100. I recommend sorting out the ideas.

We have revised this section to avoid unnecessary repetition and to eliminate any confusion regarding the number of participating farms.

Line 202.- were the assumptions of normality (normal distribution and homogeneity of variances) evaluated in the residuals of the response variable? Also, I consider that the “P-value of ≤0.10” is changed to a “P-Value of ≤0.05”. It is too high this value. Therefore, change the significance level in the results section.

We confirm that the assumptions of normality were assessed for the residuals of the response variable. Specifically, we used both the skewness and kurtosis test (sktest) in Stata, and we additionally verified normality through visual inspection of Q-Q plots and residual-versus-fitted plots.

In response to your suggestion regarding the significance threshold, we have adjusted the significance threshold from P-value ≤ 0.10 to the more conventional P-value ≤ 0.05. This revision has been applied consistently throughout both the ‘Methods’ and ‘Results’ sections .

RESULTS

General comments:

I recommend restructuring this entire section with a significance of “P-Value of ≤0.05”.

Following your recommendation, we have restructured the relevant section to consistently apply a significance threshold of P-value ≤ 0.05 throughout the manuscript.

Line 248-. They mention “The weight distribution of kids on farm B tended to be skewed toward lower weights while the weight distribution on farm C tended to be skewed toward higher weights (Figure 1).” Fig 1, says “Figure 1. Graphical representation of data collected by Dutch dairy goat farmers for each individual kid between June and August 2020”. They are referring to fig 2? Also, they mention “weight distribution on farm C tended”, specify what they mean by this.

Thank you for pointing this out. You are correct that the reference should have been to Figure 2, not Figure 1. This has  been corrected in the revised manuscript.

In addition, we have clarified the description of the weight distribution for farm C. The revised sentence now reads:
The distribution of kid weights at farm B was skewed towards lower values, while kid weights at farm C showed a tendency towards higher weight values (Figure 2).

Line 251.- Specify the statistical addresses by “Brith weight” and “Farm”.
Thank you for pointing this out. We have clarified the statistical analysis in the revised manuscript by specifying that birth weight was compared between kids across farms, and that farm was included as a factor in this analysis.

The revised sentence now reads:
“Distribution of recorded birth weights in kg for farms A, B, and C. Median birth weight differed significantly between farms (p < 0.05); see Table 4 for detailed statistical associations.

Line 271.- Table 2, the “Mean” is decimated, I recommend mentioning “Mean±DE”. This should also be adjusted in table 3.

Thank you for your suggestion. We were initially unsure whether the reviewer referred to presenting the standard deviation (SD) or the standard error (SE). To clarify, standard deviations were not included in the tables because not all parameters were normally distributed. We considered it inappropriate to present SDs for data that did not meet the assumptions of normality. Instead we decided to include the 5 and 95th percentile values for all evaluated parameters as measure of variability. If the reviewer prefers, we are willing to reconsider and include SD values for parameters that do meet normally assumptions.

Line 313.- mention “Birth weight was positively associated with weight gain”. Specify how these data were analyzed in the “Data analysis” section.

Thank you for your comment. We have clarified in the analysis section how the association between birth weight and weight gain was assessed. Specifically, we used univariate multilevel linear regression models with a random effect for farm to account for clustering within farms. Birth weight was included as an independent variable, and weight gain during the rearing period was used as the dependent variable.

DISCUSSION

In general, I recommend restructuring this section and explaining the findings observed in the research and trying to explain the mechanism involved.

We appreciate your suggestion to restructure the ‘Discussion’ section and elaborate on the mechanisms underlying the observed findings. In response, we have revised this section to provide a clearer interpretation of the results and to explain the potential biological and management-related mechanisms that may account for the observed differences between farms. For example, we now discuss how variations in colostrum quality, feeding methods, and weaning strategies could have influenced weight gain during different rearing phases.

Line 336-347.- support this paragraph with references.

We have included additional references that support the statements in these paragraphs (line 342).

Line 357-362.- support this paragraph with references.

We have included additional references that support the statements in these paragraphs (line 348).

Line 424-426.- mention “In other ruminant species, parity has been associated with colostrum IgG concentration [28], [35], and birth weight [36]”. You did not evaluate the “IgG” so you cannot relate it because it can be speculative. I recommend just focusing on the results of your research.

Thank you for your comment. In response to your observation we have removed this section from the manuscript to ensure that the discussion remains strictly focused on our own research findings and does not include speculative interpretations.

Reviewer 2 Report

Comments and Suggestions for Authors

This study ended in August 2020 – are there any further data?  Data from more farms over several years would be much more relevant and applicable than three of five farms studied over one season.  Visits by the authors on key recording days would address many of this reviewer’s concerns regarding missing data.  There are also concerns regarding the veracity of the reported mortality rates.

Lines 20-21

“Despite these challenges, farmers acknowledged the importance of high quality data for effective kid-rearing management” yet only three of the five enlisted farmers collected data and less than half the number of mating weights were recorded relative to birth weights. Line 99 refers to intensive cooperation – this did not appear to result? 

Recording weaning and mating weights are basic management data and should be a simple procedure taking one hour per 100 goats and undertaken on only one or two days.  I fail to appreciate how these data were not recorded especially given the major objective of this study.  Why did the authors not visit the goat farms on pre-determined days to record and verify the data?  If these simple measurements were not consistently recorded then how reliable are more critical data such as mortality rates. Mating weights are only important and relevant if the animals become pregnant.

Specific comments

Line 123

final wording to sentence

Line 160

Why were mortalities not investigated.  Morbidity rates would also have been relevant.

Page 6 Table 1

This table is difficult to follow – the registered mortality should also be included.

How reliable are the mortality data because the number of deaths after the 21 days is very low indeed?

Page 7

Birthweight in relation to litter size?  Are does scanned for number of foetuses – if not, why not?

Table 2

Why was passive antibody transfer not checked?  Table 2 contains a large amount of data but none of this is important unless the colostrum practice is effective – sampling for total solids in kid sera after 24 hours would be a simple and inexpensive test and is the most used by field veterinarians investigating perinatal lamb mortality.

Line 290

Presumably this reduction in average daily liveweight gain was due to nutrition? Was the possibility of disease, such as coccidiosis, considered?

How reliable are the mortality data  - were all deaths recorded? In Table 1 there is a large discrepancy between birthweight number and mating number.  I would be more convinced of the veracity of these mortality data if these other records were complete and this discrepancy was much closer to the mortality percentage.  How often was the primary investigator present on farm?  See paragraph beginning line 444.

Lines 317-9

Colostrum administration not intake or uptake.  Total solids recorded in kid sera samples after 24 hours-old would have been a more useful parameter to assess/monitor colostrum management.  See lines 426-7. Briefly mentioned in line 323 – assumes casual reader is familiar with the Brix refractometer method.

Line 356

Disease events were not monitored in this study.

Line 383

Were data recorded for kids based upon litter size?  Surely, determining foetal number and feeding accordingly is a basic flock management strategy?

Line 460

Development in technologies cannot replace someone on the farm to record key data – farmers cannot be relied upon to always record key data.  I would respectfully suggest that mortality data fall into this incomplete record.

Author Response

Response to Reviewer 2

We sincerely thank you for your valuable suggestions and constructive feedback on our manuscript. Your comments highlighted that the aim of the article was not always clearly articulated and that structural improvements were necessary to better guide the reader through the objectives of the study.

In response, we have undertaken substantial revisions to enhance the clarity, coherence, and logical flow of the manuscript. Particular attention was given to clearly stating the study's aims to develop and evaluate key performance indicators (KPIs) to assess and optimise kid-rearing practices on dairy goat farms, and restructuring the manuscript to better align with and support this objective. To strengthen the narrative, we moved outcomes from the development phase, originally included in the Materials and Methods section, to the Results section. This adjustment clarifies the progression through the study’s three key phases: development, proof of principle in practise, and evaluation, and illustrates how each contributes to the overarching research aim. Additionally, the entire manuscript has been thoroughly reviewed and revised for grammar and language to improve overall readability and consistency.

We are very grateful for the time and effort you have taken to review our manuscript. Please find our detailed responses below, with the corresponding revisions and corrections highlighted in the re-submitted files using track changes.

Point-by-point response to Comments and Suggestions for Authors

This study ended in August 2020 – are there any further data?  Data from more farms over several years would be much more relevant and applicable than three of five farms studied over one season.  Visits by the authors on key recording days would address many of this reviewer’s concerns regarding missing data.  There are also concerns regarding the veracity of the reported mortality rates.

Lines 20-21

“Despite these challenges, farmers acknowledged the importance of high quality data for effective kid-rearing management” yet only three of the five enlisted farmers collected data and less than half the number of mating weights were recorded relative to birth weights. Line 99 refers to intensive cooperation – this did not appear to result? 

Recording weaning and mating weights are basic management data and should be a simple procedure taking one hour per 100 goats and undertaken on only one or two days.  I fail to appreciate how these data were not recorded especially given the major objective of this study.  Why did the authors not visit the goat farms on pre-determined days to record and verify the data?  If these simple measurements were not consistently recorded then how reliable are more critical data such as mortality rates. Mating weights are only important and relevant if the animals become pregnant.

We thank the reviewer for this important observation. We fully agree that longitudinal data from a larger number of farms and over multiple years would strengthen the applicability and generalisability of the findings. However, the current study aimed to develop a set of key parameters to evaluate the quality of kid rearing and to test their feasibility in a real life setting. It was explicitly designed as a proof of principle study limited to a single rearing season and a small, voluntarily participating group of commercial farms.

We acknowledge the challenges encountered in on-farm data collection, including missing data and low participation from two of the five enrolled farms. Rather than being unexpected, these challenges were a key outcome of the study itself, demonstrating that even basic performance data are often not consistently recorded under routine conditions. This underlines the need for improved, user-friendly recording tools and better support systems to enhance on-farm data collection. While farm visits by the research team during key recording moments might have reduced missing data, such intensive support was intentionally excluded from the study design. Our aim was to assess whether farmers could independently integrate data recording into their routine workflows. This farmer-led approach better reflects real-world conditions, but indeed led to variation in data completeness. Additionally, farmers were not always aware of missing data and did request additional support in those instances.

Regarding the mortality rates, we fully agree that underreporting is likely, particularly of stillbirths and pre-tagging losses. This has been explicitly acknowledged in the revised discussion section (lines 431-445). It is worth noting that since November 2020, Dutch legislation requires registration of all kids within seven days postpartum, including stillbirths and sex identification. This policy change will greatly improve data completeness and accuracy in future research.

We hope this response this response clarifies the rationale behind our design choices and highlights the relevance of our findings for informing future, more extensive field investigations.

Specific comments

Line 123

final wording to sentence

This sentence has been adjusted.

Line 160

Why were mortalities not investigated.  Morbidity rates would also have been relevant.

Morbidity data were not collected in this study. During the design phase, farmers indicated that recording detailed health events for each individual kid would be difficult to implement consistently and would significantly increase the registration burden. Given the high labour demands during the kidding and rearing period, we intentionally opted to keep the data collection process as streamlined as possible. The study was therefore limited to testing the feasibility of collecting a relevant set of key indicators: birth weight, colostrum intake, growth, and mortality.

That said, we fully agree that morbidity and cause-specific mortality are highly relevant for a more comprehensive evaluating of kid rearing outcomes. These aspects should be prioritised in future studies, particularly now that digital registration systems and new national requirements are facilitating more complete data collection at the farm level.

Page 6 Table 1

This table is difficult to follow – the registered mortality should also be included.

Line 356

Disease events were not monitored in this study.

We appreciate the reviewer’s observation regarding the absence of disease monitoring and the importance of clear reporting of mortality. As noted, disease events were not tracked in this study due to the challenges of consistent health event recording under commercial farm conditions. Regarding Table 1, we would like to clarify that it presents the number of mortality events during two distinct periods (birth–21 days and birth–91 days). This table is intended to reflect the completeness of mortality data collection at each farm.

How reliable are the mortality data because the number of deaths after the 21 days is very low indeed?

We acknowledge the reviewer’s valid concern regarding the low number of recorded deaths after 21 days and the implications this has for data reliability. The observed low mortality figures, particularly beyond the early rearing period, likely reflect a combination of actual mortality patterns and underreporting.

We have addressed this limitation in the revised Discussion section, where we highlight the potential for underestimation of mortality and to stress the need for more rigorous monitoring.

Page 7

Birthweight in relation to litter size?  Are does scanned for number of foetuses – if not, why not?

Line 383

Were data recorded for kids based upon litter size?  Surely, determining foetal number and feeding accordingly is a basic flock management strategy?

Thank you for this insightful comment. Birth weight was indeed recorded in relation to litter size, and this variable was included in the dataset. However, statistical associations between birth weight and litter size were not explored in-depth in the current analysis, as the primary objective  was to assess the feasibility of on-farm data collection and to identify key performance indicators that are both relevant and practical for farmers. We fully agree that the relationship between litter size and birth weight is important for understanding intra-litter variation and early kid development.

With regard to foetal scanning: routine ultrasound scanning on litter size is not yet a standard practice on Dutch dairy goat farms, and was not performed on the participating farms. We agree that foetal scanning can provide an important basis for tailored nutrition and rearing strategies and recommend its consideration in future studies aiming to link prenatal management with birth and growth outcomes.

 Table 2

Why was passive antibody transfer not checked?  Table 2 contains a large amount of data but none of this is important unless the colostrum practice is effective – sampling for total solids in kid sera after 24 hours would be a simple and inexpensive test and is the most used by field veterinarians investigating perinatal lamb mortality.

Lines 317-9

Colostrum administration not intake or uptake.  Total solids recorded in kid sera samples after 24 hours-old would have been a more useful parameter to assess/monitor colostrum management.  See lines 426-7. Briefly mentioned in line 323 – assumes casual reader is familiar with the Brix refractometer method.

Lines 317-9

Colostrum administration not intake or uptake.  Total solids recorded in kid sera samples after 24 hours-old would have been a more useful parameter to assess/monitor colostrum management.  See lines 426-7. Briefly mentioned in line 323 – assumes casual reader is familiar with the Brix refractometer method.

Thank you for your valuable feedback. We fully agree that evaluation of passive transfer of immunity would have significantly enhanced the assessment of colostrum management practices. However, serum sampling from kids was not included in the study protocol, as it would have required blood sampling within the first days postpartum. Under Dutch legislation (Wet op de Dierproeven) such sampling classifies the study as an animal experiment, thus requiring formal ethical approval and licensing, which fell outside the scope and resources of this proof-of-principle project.

Given this limitation, we relied on data that could be collected non-invasively by the farmer or local practitioner. In practice, IgG levels in goat kids are typically determined in a limited sample and, as a result, such data were not available consistently across all study farms.

To still gain insight into the quality of colostrum fed to the kids, we opted for an accessible, farmer-implemented method using Brix refractometry to assess total solids in colostrum prior to administration. While we agree that “colostrum administration” is a more accurate term than “intake,” we used the Brix method as a widely accepted field proxy for colostrum quality.

Line 290

Presumably this reduction in average daily liveweight gain was due to nutrition? Was the possibility of disease, such as coccidiosis, considered?

Thank you for your insightful comment regarding the observed reduction in average daily liveweight gain post-weaning. We agree that nutrition is a likely contributing factor; however, disease such as coccidiosis could indeed also have played a role. We did not explicitly investigate the presence of coccidiosis in this study. However, we would like to note that coccidiosis in young ruminants is generally considered a management-related disease, resulting from an imbalance between host immunity and environmental infection pressure. Management interventions such as the timing and method of weaning can strongly influence the development of gut immunity and, consequently, the susceptibility to coccidial infection. In this context, a sudden change in diet or housing, as seen with abrupt weaning practices, may increase the risk of clinical or subclinical coccidiosis and thus contribute to growth reduction. Therefore, while coccidiosis may very well have been present, we regard it more as a potential consequence of the applied management strategy rather than an isolated explanatory factor. For this reason, it was not specifically identified or reported in the manuscript. In addition, when discussing the results, farmers did not mention notable outbreaks of coccidiosis.

How reliable are the mortality data  - were all deaths recorded? In Table 1 there is a large discrepancy between birthweight number and mating number.  I would be more convinced of the veracity of these mortality data if these other records were complete and this discrepancy was much closer to the mortality percentage.  How often was the primary investigator present on farm?  See paragraph beginning line 444.

As addressed in our earlier responses (see comments related to lines 202, 248, and 444), we acknowledge that not all deaths were recorded comprehensively across all farms. In particular, stillbirths and deaths occurring before ear-tagging, which typically takes place after the first few days, were not consistently registered, contributing to an underestimation of early mortality. Moreover, farmers were not always aware when their data were incomplete and therefore did not request assistance during the recording period.

We agree that the discrepancy between the number of birth records and mating weights indicates unrecorded losses and/or missing data. However, part of this discrepancy is also attributable to farm specific management practices. On some farms, for example, the buck kids were recorded at birth, but removed from the farm before reaching mating age, and thus were not weighed later. The study was designed as a proof of principle, embedded in a farm-based context to evaluate the feasibility of collecting individual-level kid-rearing data under routine commercial conditions. Consequently, the research team was not present during each recording moment, and on-farm visits by the primary investigator were limited. We have clarified this in the revised Discussion section and now explicitly state that the reported mortality figures likely represent a conservative estimate. We have also emphasised the need for improved, structured and ideally automated data recording methods in future studies to enhance data reliability.

Line 460

Development in technologies cannot replace someone on the farm to record key data – farmers cannot be relied upon to always record key data.  I would respectfully suggest that mortality data fall into this incomplete record.

We sincerely appreciate your observation regarding the limitations of relying solely on farmers for data collection. We agree that key outcome data, such as mortality, are at risk of being underreported when no independent recording or verification is in place. As we have acknowledged in our previous responses, and clarified in the revised manuscript, the mortality data collected in our study are likely incomplete, particularly for early losses and animals not ear-tagged. This limitation is inherent to the design of this proof-of-principle study, which aimed to assess the feasibility of farmer-led data collection under real-world conditions.

While we agree that technological solutions alone cannot fully replace the role of dedicated data recorders on farm, we believe they can help reduce the burden on farmers and minimise omissions, especially during labour-intensive periods such as kidding. We have revised the discussion to reflect this important nuance and emphasise that future studies would benefit from more direct researcher involvement or automated recording systems to improve the accuracy and completeness of key performance data.

Once again, we thank the reviewer for this critical insight, which has helped us strengthen both the interpretation and presentation of our findings.

Reviewer 3 Report

Comments and Suggestions for Authors

The manuscript is penetrating and innovative in its evaluation of the value and usefulness of individual-level data on dairy goat kids with a view to optimizing rearing practices in the Dutch dairy goat industry. The authors are outright in articulating the importance of early life and its impact on animal welfare and performance. Not only does the study gather statistically valid rearing parameters through farmer involvement, but practical relevance in a day-to-day scenario is also ensured. Inclusion of multifaceted data ranging right from birth weight and colostrum management to subsequent growth and cause-specific death is evidence of an integrated approach by providing handy benchmarks for practice in the future.
Aspects to Improve
Methodological Precision and Standardization of Data
While the study does an excellent job of establishing significant indicators of rearing, it would be made more reproducible and rigorous if it better outlined data-collecting procedures—namely, how inconsistencies on farms (such as inconsistencies in when weights were taken or colostrum recorded) were addressed. Enlarging this section by describing the procedures used to normalize data-collecting or by documenting methods to handle missing or inconsistent data would make the study more reproducible and rigorous overall.
Integration of Digital Tools into Future Applications
The discussion about the application possibility of decision support tools is promising. To make this more worthwhile, however, it would be important to include more specifics or preliminary ideas on the ways these technologies could be applied to daily operations on the farm. As an example, the article could provide some suggested layout options or testing plans for a smartphone application that could automate recording and reduce the farmer's burden in labor. Such an addition would place the applied reality value of the study into context and serve as direction on further work to be undertaken.

Author Response

Response to Reviewer 3

We would like to sincerely thank you for your thoughtful and positive assessment of our manuscript. We greatly appreciate your recognition of the study’s integrated approach and its potential contribution to optimising kid-rearing practices in the dairy goat sector. Your encouraging feedback regarding the practical relevance and innovation of the work is very motivating. We also thank you for the valuable suggestions to further improve the manuscript.

In response, we have undertaken substantial revisions to enhance the clarity, coherence, and logical flow of the manuscript. Particular attention was given to clearly stating the study's aims to develop and evaluate key performance indicators (KPIs) to assess and optimise kid-rearing practices on dairy goat farms, and restructuring the manuscript to better align with and support this objective. To strengthen the narrative, we moved outcomes from the development phase, originally included in the Materials and Methods section, to the Results section. This adjustment clarifies the progression through the study’s three key phases: development, proof of principle in practise, and evaluation, and illustrates how each contributes to the overarching research aim. Additionally, the entire manuscript has been thoroughly reviewed and revised for grammar and language to improve overall readability and consistency.

We are very grateful for the time and effort you have taken to review our manuscript. Please find our detailed responses below, with the corresponding revisions and corrections highlighted in the re-submitted files using track changes.

Kind regards,
The authors

Point-by-point response to Comments and Suggestions for Authors

The manuscript is penetrating and innovative in its evaluation of the value and usefulness of individual-level data on dairy goat kids with a view to optimizing rearing practices in the Dutch dairy goat industry. The authors are outright in articulating the importance of early life and its impact on animal welfare and performance. Not only does the study gather statistically valid rearing parameters through farmer involvement, but practical relevance in a day-to-day scenario is also ensured. Inclusion of multifaceted data ranging right from birth weight and colostrum management to subsequent growth and cause-specific death is evidence of an integrated approach by providing handy benchmarks for practice in the future.

We sincerely thank the reviewer for the thoughtful and encouraging feedback on our manuscript, as well as the valuable suggestions for improvement. We appreciate the recognition of our integrated approach and the practical relevance of our findings, and we agree with the importance of further strengthening both the methodological clarity and the applied value of the study.

Aspects to Improve
Methodological Precision and Standardization of Data

While the study does an excellent job of establishing significant indicators of rearing, it would be made more reproducible and rigorous if it better outlined data-collecting procedures—namely, how inconsistencies on farms (such as inconsistencies in when weights were taken or colostrum recorded) were addressed. Enlarging this section by describing the procedures used to normalize data-collecting or by documenting methods to handle missing or inconsistent data would make the study more reproducible and rigorous overall.

In response to your helpful comments, we have expanded the ‘Discussion’ section to more clearly address the challenges related to data consistency and standardisation across farms. Specifically, we now describe in greater detail how variation in data collection (e.g., weighing moments, colostrum recording) was managed, and how incomplete or inconsistent data were handled during analysis. These additions help to contextualise the limitations inherent to on-farm research while reinforcing the relevance of developing more standardised protocols in future studies.

“A further challenge identified in this study was inconsistency in data collection. Not all farmers recorded every variable as agreed upon during the workshops, often reflecting their individual assessment of the perceived added value against the effort required. To support broader implementation in future initiatives, it will be essential to develop clearly defined, standardised, and practically applicable data collection protocols. While intensive support during this study was valuable in facilitating adoption, such guidance is unlikely to be sustainable at scale. Ensuring that data collection methods are reproducible and rigorous, yet straightforward to implement independently by farmers, is therefore essential.”

In addition, we have discussed how farmer engagement influenced the completeness and quality of data recording, particularly for variables such as mortality. By doing so, we aim to provide greater methodological transparency and highlight critical lessons for improving study design and data reliability in future, larger-scale studies.

Integration of Digital Tools into Future Applications
The discussion about the application possibility of decision support tools is promising. To make this more worthwhile, however, it would be important to include more specifics or preliminary ideas on the ways these technologies could be applied to daily operations on the farm. As an example, the article could provide some suggested layout options or testing plans for a smartphone application that could automate recording and reduce the farmer's burden in labor. Such an addition would place the applied reality value of the study into context and serve as direction on further work to be undertaken.

We have also revised and expanded the final part of the ‘Discussion’ section to include more concrete ideas regarding the development of digital tools. In particular, we provide suggestions on how a tiered digital solution might function, from a basic version relying on routinely collected data to a more advanced option for highly engaged farmers. Additionally, we describe the potential features of a smartphone application, including user-friendly data entry formats and automated data collection (e.g., via ear tag scanners), and how these tools could reduce workload while increasing data accuracy and value to management.

We hope these additions adequately address your suggestions and further enhance the practical impact and reproducibility of the study. Thank you again for your constructive feedback.

Reviewer 4 Report

Comments and Suggestions for Authors

This is a case study and not a research paper. This study is basically just a comparison of data collected from three dairy goat farms. There is no further purpose or precise investigation behind the survey.

There are many methodology errors in the text, starting with the 5 farms mentioned in the summary which are finally only three. Some of the indicators are investigated only on one farm (e.g. Brix value). The numbers of the animals are not clear and used different numbers in the text and tables. At the begginning, female and male kids are mentioned together (of cours, litter size is very imortant regarding the growth performance), but later, male kids "are moved from the farms" and only the development of female kids is followed-up.

The data collection took place over 5 years ago. The management and housing system of the farms are not presented. There is nothing about the digital solutions in the text which can be recommended for the farmers.

Other detailed comments are in the text.

Author Response

Response to Reviewer 4

We would like to sincerely thank you for your valuable suggestions and constructive feedback on our manuscript. Your comments have helped us to recognise that the aim of the article was not always clearly articulated, and that structural changes were necessary to better guide the reader in understanding the objectives of the paper.

In response, we have undertaken substantial revisions to enhance the clarity, coherence, and logical flow of the manuscript. Particular attention was given to clearly stating the study's aims to develop and evaluate key performance indicators (KPIs) to assess and optimise kid-rearing practices on dairy goat farms, and restructuring the manuscript to better align with and support this objective. To strengthen the narrative, we moved outcomes from the development phase, originally included in the Materials and Methods section, to the Results section. This adjustment clarifies the progression through the study’s three key phases: development, proof of principle in practise, and evaluation, and illustrates how each contributes to the overarching research aim. Additionally, the entire manuscript has been thoroughly reviewed and revised for grammar and language to improve overall readability and consistency.

We are very grateful for the time and effort you have taken to review our manuscript. Please find our detailed responses below, with the corresponding revisions and corrections highlighted in the re-submitted files using track changes.

Kind regards,
The authors

Point-by-point response to Comments and Suggestions for Authors

This is a case study and not a research paper. This study is basically just a comparison of data collected from three dairy goat farms. There is no further purpose or precise investigation behind the survey.

Although we understand the concern regarding the scope of our study, we respectfully disagree with the characterisation of the study as merely a case study without further purpose. While the study was conducted on a limited number of farms, it was explicitly designed to develop a set of key performance indicators to evaluate the quality of kid rearing. The study was structured as a proof-of-principle investigation, aimed at assessing the feasibility, practical applicability and added value of individual-level KPIs for kid rearing in real-life dairy goat farming. The study follows a structured research design, comprising: a participatory development phase to define relevant KPIs; systematic data collection on-farm using these KPIs; statistical analysis to explore associations between management indicators and kid growth performance; and a critical evaluation of the feasibility and quality of on-farm data collection.

The outcomes provide not only insight in the added value of the collected data, but also revealed challenges in collection of the requested data resulting in incompleteness and a lack of data quality in two of the farms. These insights are essential for informing the development of more robust recording systems and decision support tools in the future.

We believe that the structured methodology, clearly defined objectives, and contribution to both scientific knowledge and practical application, distinguish this work from a simple case comparison, and we hope that the revisions made further clarify the scientific relevance and purpose of the study.

There are many methodology errors in the text, starting with the 5 farms mentioned in the summary which are finally only three. Some of the indicators are investigated only on one farm (e.g. Brix value). The numbers of the animals are not clear and used different numbers in the text and tables. At the begginning, female and male kids are mentioned together (of cours, litter size is very imortant regarding the growth performance), but later, male kids "are moved from the farms" and only the development of female kids is followed-up.

The data collection took place over 5 years ago. The management and housing system of the farms are not presented. There is nothing about the digital solutions in the text which can be recommended for the farmers.

Other detailed comments are in the text. https://susy.mdpi.com/user/review/displayFile/58568480/al0oOLKY?file=review&report=45584929

We thank the reviewer for these important observations. We appreciate the opportunity to clarify these points and have made several revisions to the manuscript to address them. We fully agree that longitudinal data from a larger number of farms and over multiple years would strengthen the  generalisability of the findings. However, the current study was intentionally designed as a proof-of-principle project, aimed to develop a set of key parameters to evaluate the quality of kid rearing, and to test these parameters under commercial farm conditions. It was limited to a single rearing season and a small cohort of voluntarily participating farms, focusing primarily on assessing the feasibility of farmer-led data collection in real-world settings.

Regarding the number of farms:
Although five farms initially enrolled, two farms were unable to complete all phases of data collection. This is now clearly explained in the Materials and Methods and Results sections. The challenges in maintaining full participation reflect one of the key findings of the study, reinforcing the importance of developing more supportive and standardised tools for future on-farm data recording.

Regarding the use of specific indicators such as Brix values, we acknowledge that not all indicators were recorded consistently on all farms. This limitation is explicitly discussed in the revised manuscript, and we have clearly indicated in the Results section where data were farm-specific. This variation reflects the real-life conditions under which the feasibility of data collection was evaluated.

The manuscript has been carefully revised to improve consistency in the reporting of animal numbers between the text and tables. Discrepancies occurred because not all variables were consistently recorded for every animal throughout the study. This variation itself highlights the feasibility challenges of farmer-led data collection and further underlines the need for standardisation. Both male and female kids were included in the perinatal analysis. However, since most male kids were moved off-farm shortly after birth, only female kids (doelings) were included in the postnatal and later growth analyses. This is clarified in the Methods and Results section of the revised paper to avoid misunderstanding.

As advised, we have revised the Abstract to better reflect the key findings, conclusions, and implications of the study. Additionally, we have revised a paragraph in the Materials and Methods section to improve clarity and precision. Finally, we have replaced Figure 1 with a higher-resolution version to enhance visual quality and readability.

We hope that these clarifications address your concerns and emphasise how this study lays a foundation for future, more extensive investigations.